# Monocyte-Derived miRNA-1914-5p Attenuates IL-1β–Induced Monocyte Adhesion and Transmigration

**DOI:** 10.3390/ijms24032829

**Published:** 2023-02-01

**Authors:** Kohki Toriuchi, Toshie Kihara, Hiromasa Aoki, Hiroki Kakita, Satoru Takeshita, Hiroko Ueda, Yasumichi Inoue, Hidetoshi Hayashi, Yohei Shimono, Yasumasa Yamada, Mineyoshi Aoyama

**Affiliations:** 1Department of Pathobiology, Nagoya City University Graduate School of Pharmaceutical Sciences, 3-1 Tanabe-dori, Mizuho-ku, Nagoya 467-8603, Japan; 2Department of Perinatal and Neonatal Medicine, Aichi Medical University, 1-1 Yazakokarimata, Nagakute 480-1195, Japan; 3Department of Cell Signaling, Nagoya City University Graduate School of Pharmaceutical Sciences, 3-1 Tanabe-dori, Mizuho-ku, Nagoya 467-8603, Japan; 4Department of Innovative Therapeutic Sciences, Cooperative Major in Nanopharmaceutical Sciences, Nagoya City University Graduate School of Pharmaceutical Sciences, 3-1 Tanabe-dori, Mizuho-ku, Nagoya 467-8603, Japan; 5Department of Biochemistry, Fujita Health University School of Medicine, 1-98 Dengakugakubo, Kutsukake-cho, Toyoake 470-1192, Japan

**Keywords:** atherosclerosis, monocyte adhesion, monocyte transmigration, miR-1914-5p, Mac-1, MCP-1

## Abstract

Atherosclerosis can lead to cardiovascular and cerebrovascular diseases. Atherosclerotic plaque formation is promoted by the accumulation of inflammatory cells. Therefore, modulating monocyte recruitment represents a potential therapeutic strategy. In an inflammatory state, the expression of adhesion molecules such as intercellular adhesion molecule-1 (ICAM-1) is upregulated in endothelial cells. We previously reported that miR-1914-5p in endothelial cells suppresses interleukin (IL)-1β–induced ICAM-1 expression and monocyte adhesion to endothelial cells. However, whether monocyte miR-1914-5p affects monocyte recruitment is unclear. In this study, IL-1β decreased miR-1914-5p expression in a human monocyte cell line. Moreover, miR-1914-5p inhibition enhanced adhesion to endothelial cells with the upregulation of macrophage-1 antigen (Mac-1), a counter-ligand to ICAM-1. Transmigration through the endothelial layer was also promoted with the upregulation of monocyte chemotactic protein-1 (MCP-1). Furthermore, a miR-1914-5p mimic suppressed IL-1β–induced monocyte adhesion and transmigration in monocytes with Mac-1 and MCP-1 downregulation. Further investigation of miR-1914-5p in monocytes could lead to the development of novel diagnostic markers and therapeutic strategies for atherosclerosis.

## 1. Introduction

Atherosclerosis can lead to cardiovascular and cerebrovascular diseases through the formation of lesions characterized by the accumulation of inflammatory cells in vascular walls [1]. Circulating monocytes attach to vessel walls and migrate into the arterial intima in response to chemotactic factors and cytokines produced by vascular endothelial cells, smooth muscle cells (SMCs), and infiltrated cells [2]. These invading monocytes then differentiate into macrophages and internalize low-density lipoprotein (LDL), resulting in the formation of foam cells that amplify the inflammatory response via pro-inflammatory cytokine and chemokine release [3]. Therefore, modulating the recruitment of monocytes represents a potential therapeutic approach [4].

Monocyte trafficking is regulated by interaction with activated endothelial cells, which is coordinated by endothelial cell adhesion molecules, chemokines, and their receptors [5,6]. Inflammatory cytokines and oxidative LDL upregulate the expression of cell adhesion molecules such as intercellular adhesion molecule-1 (ICAM-1) on activated endothelial cells [7,8]. ICAM-1 promotes monocyte adhesion by binding to the counter-ligand macrophage-1 antigen (Mac-1, a heterodimer consisting of CD11b and CD18), which is expressed on monocytes [9,10]. Monocytes that adhere to the blood vessel wall are arrested by monocyte chemotactic protein-1 (MCP-1) and then begin to prepare to migrate [11]. Before transmigration, monocytes cross the venular wall and enter the blood vessel in an efficient Mac-1– and ICAM-1–dependent manner [5,12,13]. During transmigration, Mac-1 may enhance inflammatory cell recruitment by interacting with a counter receptor [14]. Therefore, Mac-1 and MCP-1 are considered key factors in the recruitment of monocytes into the subendothelial zone.

MicroRNAs (miRNAs) are a class of short, non-coding RNAs of 18-22 nucleotides in length that regulate gene expression at the post-transcriptional level via messenger RNA (mRNA) degradation or translational repression [15]. In an inflammatory state, the expression of adhesion molecules such as ICAM-1, vascular cell adhesion molecule-1, and E-selectin is regulated by several miRNAs [16]. We previously reported that expression of the miRNA miR-1914-5p is suppressed by IL-1β, an important regulator mediating the expression of ICAM-1 in endothelial cells [17]. However, whether IL-1β affects the expression of miR-1914-5p in monocytes or the regulation of monocyte recruitment via adhesion molecules or chemokine expression remains unclear.

In the present study, we investigated whether IL-1β suppresses miR-1914-5p expression in monocytes. In addition, we examined the effect of miR-1914-5p on monocyte adhesion and transmigration.

## 2. Results

### 2.1. IL-1β Exposure Enhanced Human Monocyte Mac-1 Expression and Adhesion to Human Endothelial Cells

We previously reported that miR-1914-5p in endothelial cells suppresses IL-1β–induced ICAM-1 expression and monocyte adhesion to endothelial cells. However, whether IL-1β stimulation affects monocyte recruitment via changes in adhesion molecule expression in monocytes remains unclear. We therefore assessed Mac-1 expression by THP-1 human monocytes using flow cytometry analysis. The proportion of THP-1 cells expressing Mac-1 (CD11b^+^, CD18^+^ cells) increased compared to control cells following IL-1β exposure (Figure 1A,B). Using a cell adhesion assay, we then evaluated the adhesion of IL-1β–exposed THP-1 cells to EA.hy926 human endothelial cells. Adhesion of IL-1β–exposed THP-1 cells to EA.hy926 cells was enhanced compared with cells not exposed to IL-1β (Figure 1C,D).

### 2.2. IL-1β Exposure Enhanced Expression of MCP-1 by Human Monocytes and Their Transmigration through a Human Endothelial Cell Layer

We next examined monocyte transmigration through the endothelium as a subsequent progressive stage in the formation of atherosclerotic lesions following adhesion. IL-1β exposure increased the accumulation of transcripts of the MCP-1 gene in THP-1 cells (Figure 1E). We then assessed the transmigration of IL-1β–exposed THP-1 cells through a layer of endothelial cells using a transmigration assay. EA.hy926 human endothelial cells were seeded in Transwell inserts at confluent density, and then fluorescently labeled THP-1 cells were added to the upper chamber and allowed to transmigrate for 24 h (Figure 1F). THP-1 cells that passed through the endothelial layer into the lower chamber were evaluated by measuring the fluorescence intensity in the lower chamber using a microplate reader. The fluorescence of the lower chamber increased in analyses of IL-1β–exposed THP-1 cells compared with control cells (Figure 1G). Furthermore, we examined the THP-1 cells that passed through the endothelial layer and adhered to the underside of the membrane using a confocal microscope. The number of THP-1 cells on the underside of the membrane was increased in samples exposed to IL-1β in comparison to the number observed in the absence of IL-1β (Figure 1H,I).

### 2.3. Expression of miR-1914-5p Was Downregulated by IL-1β Exposure in Human Monocytes; Inhibition of miR-1914-5p Expression Induced Monocyte Adhesion to Human Endothelial Cells

To determine how miR-1914-5p affects the atherosclerotic inflammatory condition, we analyzed the expression of miR-1914-5p upon exposure to IL-1β. miR-1914-5p accumulation in THP-1 human monocytes was attenuated by IL-1β exposure (Figure 2A). We then examined whether the inhibition of miR-1914-5p expression by THP-1 cells affected their adhesion to endothelial cells. Initial experiments confirmed that miR-1914-5p accumulation in THP-1 cells was suppressed by a miRNA inhibitor of human miR-1914-5p (Figure 2B). Suppression of miR-1914-5p expression in THP-1 cells increased the proportion of cells expressing Mac-1 (CD11b^+^, CD18^+^ cells), even in the absence of IL-1β (Figure 2C,D). Furthermore, a higher proportion of THP-1 cells expressing low levels of miR-1914-5p adhered to EA.hy926 cells compared with the negative control group (Figure 2E,F).

### 2.4. Inhibition of miR-1914-5p Expression in Human Monocytes Upregulated the Expression of MCP-1 and Increased Monocyte Transmigration through a Human Endothelial Cell Layer

Next, we assessed whether inhibition of miR-1914-5p expression in THP-1 cells affects their transmigration through an endothelial cell layer. THP-1 cells in which miR-1914-5p expression was suppressed exhibited increased accumulation of transcripts of the MCP-1 gene even in the absence of IL-1β (Figure 3A). Compared with the negative control, the fluorescence intensity associated with THP-1 cells expressing low levels of miR-1914-5p was increased in the lower transwell chamber, indicating enhanced transmigration of these cells (Figure 3B). Moreover, the number of THP-1 cells on the underside of the membrane was increased (Figure 3C,D).

### 2.5. Overexpression of miR-1914-5p Suppressed the Adhesion of IL-1β–Exposed Human Monocytes to Human Endothelial Cells

We also examined the effect of exogenous miR-1914-5p on adhesion of THP-1 cells to endothelial cells. Initial experiments confirmed that miR-1914-5p accumulation increased in THP-1 cells exposed to a miRNA that mimicked human miR-1914-5p (Figure 4A). The proportion of Mac-1 (CD11b^+^, CD18^+^ cells)-expressing THP-1 cells in which miR-1914-5p was overexpressed decreased, even in the presence of IL-1β (Figure 4B,C). Additionally, miR-1914-5p overexpression reduced the number of cells adhered to EA.hy926 cells, even with IL-1β exposure (Figure 4D,E).

### 2.6. Overexpression of miR-1914-5p in IL-1β–Exposed Human Monocytes Suppressed MCP-1 Expression and Transmigration through a Human Endothelial Cell Layer

We also examined whether exogenous miR-1914-5p suppressed the transmigration of THP-1 cells through an endothelial cell layer. THP-1 cells overexpressing miR-1914-5p exhibited a decrease in the accumulation of transcripts of the MCP-1 gene, even in the presence of IL-1β (Figure 5A). Furthermore, miR-1914-5p overexpression resulted in a decrease in the fluorescence intensity in the lower transwell chamber and a decrease in the number of THP-1 cells on the underside of the membrane, even with IL-1β exposure (Figure 5B–D). 

## 3. Discussion

Elevated blood LDL cholesterol levels are considered to be a major causally related risk factor for the development of atherosclerosis [18]. Oxidized LDL induces IL-1β secretion via MLRP3 inflammasome activation, and subsequent inflammatory processes are thought to accelerate plaque formation [19]. In a large clinical trial, the Canakinumab Anti-inflammatory Thrombosis Outcome Study (CANTOS), anti-inflammatory therapy against atherosclerosis was found to be effective, and IL-1β played an important role in atherosclerotic plaque generation [20]. In our present study, IL-1β increased Mac-1 expression on the membrane surface of THP-1 human monocytes and the adhesion of these monocytes to EA.hy926 human endothelial cells (Figure 1A–D). Furthermore, monocyte transmigration was promoted by IL-1β stimulation (Figure 1F–I). Under these conditions, expression of the gene encoding the major chemotactic agent MCP-1 was upregulated (Figure 1E). These results suggest that inflammation-mediated monocyte recruitment is regulated by Mac-1 and MCP-1 and that monocytes in a lesion can recruit additional monocytes to the lesion.

Although IL-1β may be an attractive therapeutic target, infection in such therapy would be of concern because of the important role of IL-1β in host defense, as reported by the CANTOS [21]. Therefore, it is necessary to elucidate the details of the mechanism underlying monocyte recruitment. In our present study, we focused on miRNAs downstream of IL-1β. Details regarding miRNA-mediated regulation of monocyte trafficking, including adhesion molecule expression and monocyte transmigration, remain poorly understood [22]. First, we evaluated whether IL-1β affects the expression of miR-1914-5p in THP-1 human monocytes and found that IL-1β decreased miR-1914-5p expression in THP-1 cells (Figure 2A). Notably, degradation of miR-1914-5p enhanced Mac-1 expression in THP-1 monocytes and their adhesion to endothelial cells (Figure 2B–F). Transmigration through an endothelial cell layer was also promoted following upregulation of MCP-1 expression (Figure 3). Furthermore, treatment with a miR-1914-5p mimic suppressed IL-1β–induced monocyte adhesion and transmigration in conjunction with Mac-1 and MCP-1 downregulation (Figure 4 and Figure 5). In endothelial cells, miR-1914-5p was shown to suppress IL-1β–induced expression of ICAM-1 [17]. miR-1914-5p is considered a dual regulator in the development of atherosclerosis, inhibiting monocyte adhesion and infiltration by regulating both monocytes and endothelial cells. Other miRNAs reportedly play roles in cell adhesion and differentiation to macrophages or foam cells [23]. An effective strategy may be to target several miRNA-related steps in plaque formation. Therefore, the effect of miR-1914-5p on any other steps during atherogenesis should be examined, including neutrophil recruitment, SMC proliferation, and foam cell formation.

In this case, miR-1914-5p targets a number of genes in monocytes and affects the biology and functionality of endothelial cells. Using a public database (http://mirdb.org: accessed on 15 May 2021), we previously identified target mRNAs with potential binding sites for miR-1914-5p [17]. Short/branched chain acyl-CoA dehydrogenase (ACADSB), a target gene of miR-1914-5p, catalyze the dehydrogenation of acyl-CoA derivatives in the metabolism of fatty acids [24]. Lipid processing in macrophages is one of the most important regulators of atherosclerosis progression [25]. In addition, another target gene RNA-binding motif, single-stranded-interacting protein 1(RBMS1) reportedly promote cell migration and invasion through IL-6 expression and JAK2/STAT3 downstream signaling [26]. Furthermore, variants of the ubiquitin-conjugating enzyme E2Z gene (UBE2Z) are significantly associated with hypertriglyceridemia [27] as well as being a possible risk factor for coronary artery disease [28]. In the present study, we confirmed that the putative target genes of miRNA-1914-5p were induced by IL-1β or the miR-1914-5p inhibitor, both of which suppressed the expression level of miRNA-1914-5p (Appendix A). To further validate the miR-1914-5p-mediated regulation of its target gene in human THP-1 monocytic cells, both wild-type and its mutated variant of the RBMS1 3′UTR were inserted downstream of a luciferase minigene in bio-luminescent reporter plasmids. The results showed that IL-1β, which suppresses the expression level of miR-1914-5p (Figure 2A), was able to enhance the luciferase activity of the constructs encoding the wild-type version of the RBMS1 3′UTR in THP-1 cells, and this effect was completely abrogated when the mutations were introduced within the target site for miR-1914-5p (Appendix A). These data suggest that miRNA-1914-5p could be a functional miRNA that directly regulates its target gene, such as RBMS1, in monocytic cells.

The stage-dependent effects of IL-1 isoforms on experimental atherosclerosis have been reported. In experimental model mice, IL-1α was shown to play an important role in arterial remodeling during the early stages of atherogenesis, whereas IL-1β was shown to promote inflammation as lesion formation progresses [29]. The results of our present study suggest that IL-1β recruits monocytes into inflammatory intima via miR-1914-5p. Therefore, miR-1914-5p may be involved in further IL-1β–induced formation of plaque.

miRNAs are potential biomarkers because they are encapsulated in exosomes and released into the bloodstream [30]. Indeed, decreased miR-1914-5p levels were reported in coronary artery disease patients [31]. It is possible that miR-1914-5p in exosomes is difficult to detect because it is present in extremely low amounts. However, monocytes are circulating cells that can be collected relatively easily by blood sampling. In this study, we revealed that monocytes exposed to IL-1β exhibit decreased miR-1914-5p expression (Figure 2A). By monitoring miR-1914-5p via circulating monocytes, it may be possible to use low miR-1914-5p expression as a biomarker in combination with other miRNAs for the early diagnosis of atherosclerosis and coronary artery disease.

Several limitations of our study must be taken into consideration. First, we used an in vitro experimental model to confirm the direct relationship between miR-1914–5p and monocyte adhesion induced by IL-1β in a human monocyte cell line. However, it is not clear whether these results are valid in atherosclerosis in vivo. Further in vivo experiments monitoring the early stages of atherogenesis are needed to confirm these in vitro results. These additional studies could include apolipoprotein E–deficient mice, low-density-lipoprotein receptor–deficient mice, and mice injected with the dominant mutant of proprotein convertase subtilisin/kexin type 9 [32]. Unfortunately, these rodent models are not currently available, because miR-1914-5p expression has been confirmed only in humans and not in rodents. Second, although neutrophils also contribute to plaque formation, we focused on monocytes in our present study [33,34,35]. miR-1914-5p deficiency–induced interactions between MAC-1 and ICAM-1 could play an important role in neutrophil adhesion to the endothelium. Monocytic MCP-1 could recruit neutrophils to lesions. Third, we did not consider adhesion factors other than MAC-1. The change in the percentage of adherent cells was larger relative to the amount of change in the percentage of monocytes expressing MAC-1. These results suggest that other adhesion molecules contribute to the effect of miR-1914-5p on monocyte recruitment. Further investigations focusing on lymphocyte function-associated antigen-1, very late antigen-4, and P-selectin glycoprotein ligand-1 on the surface of monocytes in the inflammatory state are needed [36,37,38,39].

In conclusion, miR-1914-5p was identified as an IL-1β–responsive miRNA that suppresses Mac-1 and MCP-1 expression in THP-1 human monocytes. miR-1914-5p inhibitor promoted THP-1 monocyte adhesion and transmigration through the endothelium. Furthermore, exogenous miR-1914-5p counteracted the enhanced monocyte adhesion to endothelial cells and their transmigration even with exposure to IL-1β. These findings suggest that miR-1914-5p in monocytes regulates monocyte recruitment under inflammatory conditions through vascular endothelial cells. Further investigation using an in vivo pathological model could lead to an effective new therapeutic approach that promotes miR-1914-5p expression to treat atherosclerosis.

## 4. Materials and Methods

### 4.1. Cell Culture

THP-1 human monocytes and EA.hy926 human vascular endothelial cells were purchased from the American Type Culture Collection (Manassas, VA, USA). THP-1 cells were grown in RPMI-1640 (Wako, Osaka, Japan) supplemented with 10% fetal bovine serum (FBS), 100 U mL^−1^ penicillin, and 100 μg mL^−1^ streptomycin. For IL-1β treatment experiments, THP-1 cells were plated on 35-mm tissue culture dishes at 5 × 10^5^ cells/dish and then resuspended in FBS-free medium containing the indicated concentration of IL-1β (Wako). EA.hy926 cells were grown in high-glucose (4500 mg L^−1^) Dulbecco’s Modified Eagle’s Medium (Wako) supplemented with 10% FBS, 100 U mL^−1^ penicillin, and 100 μg mL^−1^ streptomycin. All cells were cultured at 37 °C in a 5% CO_2_/95% air environment and sub-cultured every third or fourth day to prevent loss of morphological characteristics.

### 4.2. miRNA Isolation and Quantitative Reverse Transcription–Polymerase Chain Reaction (qRT-PCR)

Changes in miRNA expression were analyzed using qRT-PCR, as previously described [17]. To obtain small RNAs (<200 bp), total RNA was isolated using ISOGEN II (Nippon Gene, Toyama, Japan) according to the manufacturer’s instructions. miRNA reverse transcription (RT) was performed using a TaqMan microRNA Reverse Transcription kit (Thermo Fisher Scientific, Waltham, MA, USA) following the manufacturer’s instructions. Real-time PCR was carried out on a StepOnePlus Real-Time System (Thermo Fisher Scientific) using TaqMan Universal PCR Master Mix (Thermo Fisher Scientific) and TaqMan Micro RNA Assay containing PCR primers and probes (Thermo Fisher Scientific). Expression of each miRNA was normalized to the expression of miR-16 as the housekeeping miRNA.

### 4.3. qRT-PCR

The expression of selected genes was analyzed by qRT-PCR using a StepOnePlus Real-Time System (Thermo Fisher Scientific), as previously described with minor modifications [17]. Total RNA was extracted using RNAiso plus (Takara Bio, Inc., Otsu, Japan), and reverse transcription was performed using PrimeScript RT Master Mix (Takara Bio, Inc.). The synthesized cDNAs were subjected to PCR amplification using Go Taq (Promega Corp., Madison, WI, USA). Relative quantification of MCP-1 gene was normalized against that of the *ACTB* gene. The primer pairs used for amplification were as follows: *ACTB*: forward, 5′-GATCAAGATCATTGCTCCTCCT-3′, *ACTB*: reverse, 5′-GGGTGTAACGCAACTAAGTCA-3′; *MCP-1*: forward, 5′-GCAGCCACCTTCATTCCCCA-3′, *MCP-1*: reverse, 5′-CACAGATCTCCTTGGCCACAAT-3′, *ACADSB*: forward, 5′-TGGAATACACTTTGCTCCCCT-3′, *ACADSB*: reverse, 5′-CCATGGTTGAAACCAAAGGTG-3′; *RBMS1*: forward, 5′-GTACCCTCAGTACGCCACCT-3′, *RBMS1*: reverse, 5′-GTGGGCTGGGACCAGAGA-3′, *UBE2Z*: forward, 5′-CCACCTCGGGTCAAACTGAT-3′, *UBE2Z*: reverse, 5′-CCGGGCTCATTGTGATAGGG-3′.

### 4.4. Transfection

Cell transfections were performed using GenomONE^®^-Si (Isihara Sangyo Kaisya, Ltd., Osaka, Japan) according to the manufacturer’s instructions. Hsa-miR-1914-5p inhibitor (MISSION^®^ Synthetic miRNA Inhibitor Human hsa-miR-1914-5p) and mimic (MISSION^®^ Synthetic miRNA mimic Human hsa-miR-1914-5p) were purchased from Sigma (St. Louis, MO, USA). For the control, inhibitor-NC (MISSION^®^ Synthetic miRNA Inhibitor Negative Control 1, Sigma) or mimic-NC (MISSION^®^ Synthetic miRNA mimic Negative Control 1, Sigma) were used, respectively. THP-1 cells were transfected for 24 h with miR-1914-5p inhibitor or miR-1914-5p mimic at a final concentration of 10 nM.

### 4.5. Reporter Plasmid Construction and Luciferase Reporter Assay

The 3′ UTR of the RBMS1 mRNA including the binding site for miR-1914-5p were amplified by RT-PCR from the cDNA of THP1 cells and cloned into the pGL3-MC luciferase reporter vector [40]. The QuickChange site-directed Mutagenesis kit (Agilent Technologies, Santa Clara, CA, USA) was used to mutate the binding sites as the manufacturer’s protocol described. The THP-1 cells were cotransfected with: (i) a pGL3-MC luciferase expression construct; (ii) the pRL-TK Renilla luciferase vector (Promega) using TRANS-IT (Takara Bio), following the manufacturer’s instructions. In this case, 24 h after transfection, luciferase activity was quantified and normalized to Renilla luciferase activity, using the Dual-Luciferase Reported Assay System (Promega), following the manufacturer’s instructions.

### 4.6. Flow Cytometry

Antibodies used for fluorescence-activated cell sorting (FACS) analyses were as follows: phycoerythrin (PE)/cyanine 7 anti-human CD11b antibody, PE/cyanine 7 mouse IgG1 κ isotype control antibody, Alexa Fluor 488–conjugated anti-human CD18 antibody, and Alexa Fluor 488–conjugated mouse IgG2a κ isotype control antibody. All antibodies were obtained from BioLegend (San Diego, CA, USA). THP-1 cells were washed with phosphate-buffered saline (PBS) and treated with Human BD Fc Block (BD Biosciences, San Jose, CA, USA) for 10 min on ice to block nonspecific antibody binding. Cells were incubated for 30 min with staining medium mixed with anti-human CD11b antibody and anti-human CD18 antibody. After washing with PBS, cells were resuspended with 7-amino-actinomycin D (7-AAD) (BD Biosciences). Cell populations were analyzed using a FACS Verse system (BD Biosciences), and CD11b^+^ and CD18^+^ cells were defined by the isotype control of each dye, excluding dead cells, which were highly positive for 7-AAD.

### 4.7. Cell Adhesion Assay

Cell adhesion assays were performed as previously described [17]. THP-1 cells stimulated with IL-1β or transfected with miRNA inhibitor/mimic were suspended in 10 μM BCECF-AM (Dojindo Laboratories, Kumamoto, Japan) for 1 h at 37 °C. BCECF-AM–suspended THP-1 cells (3 × 10^5^ cells/dish) were then added to a culture of EA.hy926 cells that had been seeded at 3 × 10^5^ cells/dish in 35-mm glass-bottom culture dishes (Matsunami Glass, Inc., Ltd., Osaka, Japan). After 1 h of incubation, the adherent cells were gently washed three times with RPMI 1640 to remove non-adherent cells. Fluorescence images were obtained using an LSM800 confocal microscope (Carl Zeiss, Germany). The number of THP-1 cells per view was also determined from randomly acquired images.

### 4.8. Transmigration Assay

To assess monocyte transmigration through an EA.hy926 cell monolayer, we used the Chemotaxicell, Transwell inserts with 8.0-μm pores (KURABO, Osaka Japan). The Chemotaxicell was inserted into a 24-well plate, and the upper chamber was coated with 1 μg cm^−2^ of fibronectin (Fibronectin Solution from Human Plasma; Wako). EA.hy926 cells (7 × 10^4^ cells/well) were seeded in the upper chamber of the fibronectin-coated Chemotaxicell and cultured for 2 days to form a confluent monolayer. THP-1 cells stimulated with IL-1β or transfected with miRNA inhibitor/mimic were suspended in 10 μM BCECF-AM, and BCECF-AM–suspended THP-1 cells (7 × 10^5^ cells/well) were added to the upper chamber and allowed to migrate through the EA.hy926 cell monolayer into the lower chamber by incubation at 37 °C for 24 h. The upper chamber was then removed to stop transmigration. The fluorescence intensity of medium in the lower chamber was measured using a Nivo 3S fluorescence microplate reader (Perkin Elmer, Inc., Waltham, MA, USA). Subsequently, the cells in the upper chamber were fixed and subjected to immunostaining for vascular endothelial (VE)-cadherin. During the immunostaining protocol, the upper chamber was inserted into another 24-well plate, and the lower chamber was filled with the same solution as the upper chamber in each protocol to retain the transmigrated THP-1 cells on the underside of the membrane. The membrane was washed with PBS and then fixed with 3% paraformaldehyde in PBS for 30 min. After washing with PBS, the membrane was incubated with 3% bovine serum albumin and 0.1% glycine in PBS for 1 h to block non-specific binding and then incubated overnight at 4 °C with anti–VE-cadherin primary antibody (VE-cadherin (F-8):sc-9989, 1:25 dilution; Santa Cruz Biotechnology, Inc., Dallas, TX, USA). After washing with PBS, the membrane was incubated with Alexa Fluor 594–conjugated secondary antibody (1:1000 dilution; Invitrogen, Carlsbad, CA, USA) for 1 h in the dark. Finally, the membrane was removed from the chamber using a Membrane Cutter (KURABO) and mounted onto glass slides with SlowFade Gold antifade reagent (Invitrogen). Fluorescence images were obtained using an LSM800 confocal microscope (Carl Zeiss) while confirming the spatial location of cells on the membrane via Z-stack analysis. The number of THP-1 cells on the underside of the membrane was also determined from randomly acquired images.

### 4.9. Statistical Analysis

All statistical analyses were performed using EZR software (Saitama Medical Center, Jichi Medical University, Saitama, Japan), which is a graphical user interface for R (The R Foundation for Statistical Computing, Vienna, Austria). More precisely, EZR is a modified version of R commander designed to add statistical functions frequently used in biostatistics [41]. Continuous data were compared between three or more groups using a two-tailed analysis of variance with post hoc Bonferroni’s or Turkey’s test. Data from two groups were compared using two-tailed non-paired Student’s *t*-tests. Data are presented as the mean ± SEM. Statistical significance was set at *p* < 0.05.

## Figures and Tables

**Figure 1 ijms-24-02829-f001:**
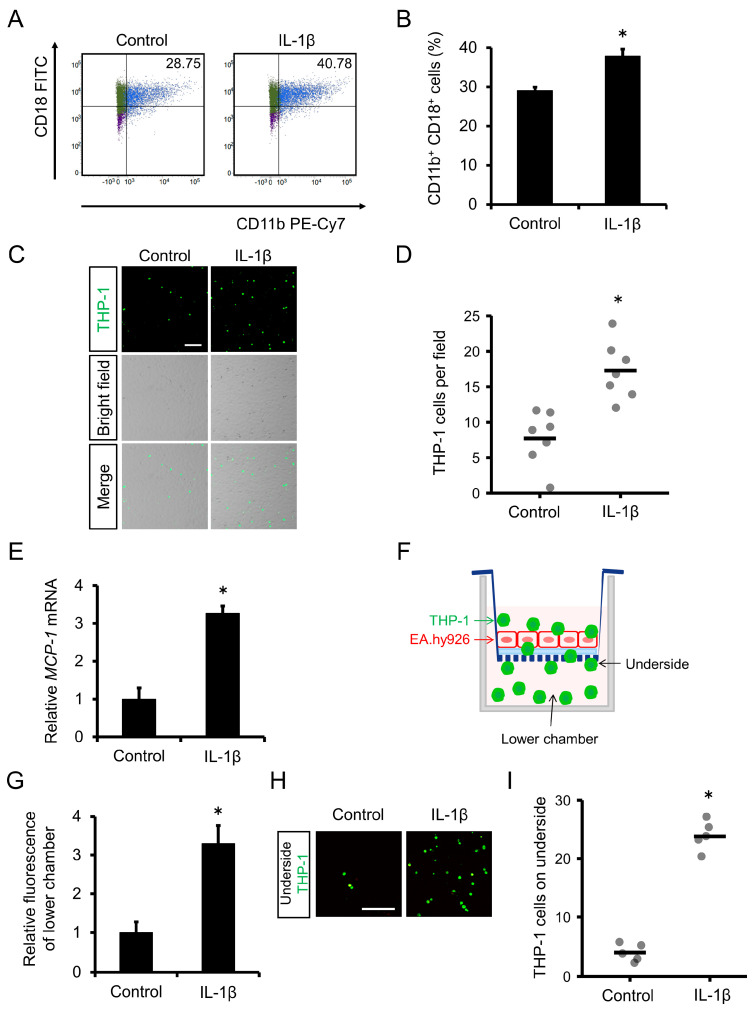
Effect of IL-1β on monocyte adhesion to endothelial cells and transmigration. (**A**) THP-1 cells were incubated with IL-1β for 24 h, and Mac-1 expression was examined by flow cytometry. THP-1 cells expressing Mac-1 were defined as CD11b^+^, CD18^+^ cells. (**B**) Proportion of cells expressing Mac-1 (CD11b^+^, CD18^+^ cells) among live cells. *n* = 3 in each group. * *p* < 0.05 compared with the control group. (**C**) Adhesion of THP-1 cells to an EA.hy926 cell monolayer. Fluorescent THP-1 cells and EA.hy926 cells were co-cultured for 1 h after treatment of THP-1 cells with IL-1β (10 ng mL^−1^). Scale bar represents 100 μm. (**D**) Number of fluorescent THP-1 monocytes on the EA.hy926 cell monolayer. *n* = 7 fields in each group. * *p* < 0.05 compared with the control group. (**E**) THP-1 cells were incubated with IL-1β for 24 h, and MCP-1 gene expression was examined by qRT-PCR. *n* = 3 in each group. * *p* < 0.05 compared with the control group. (**F**) Schematic diagram of the transmigration assay. (**G**) Transmigration of THP-1 cells through the endothelial layer into the lower chamber was assessed by comparing the relative fluorescence intensity of the upper and lower chambers using a microplate reader. *n* = 3 in each group. * *p* < 0.05 compared with the vehicle group. (**H**) THP-1 cells that passed through the endothelial layer and adhered to the underside of the membrane were detected using a confocal microscope. THP-1 cells were labeled with the fluorescent dye BCECF-AM (green). Scale bar represents 100 μm. (**I**) The number of fluorescent THP-1 monocytes on the underside of the membrane was determined. *n* = 5 fields in each group. * *p* < 0.05 compared with the control group.

**Figure 2 ijms-24-02829-f002:**
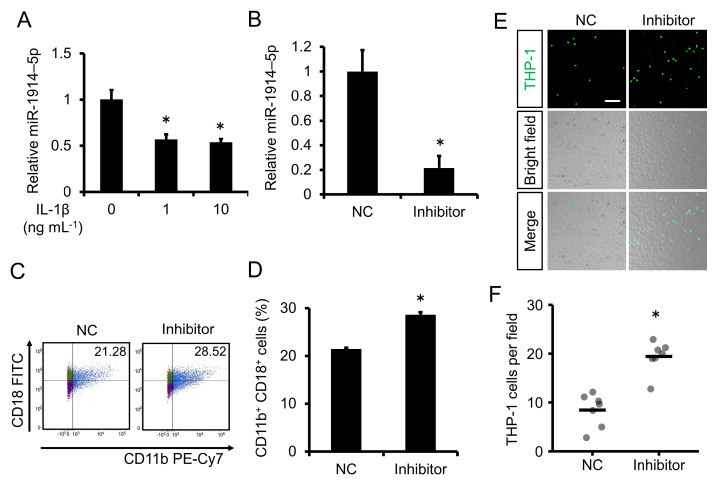
IL-1β decreased miR-1914-5p expression in monocytes, and transfection with a miR-1914-5p inhibitor increased monocyte adhesion to endothelial cells. (**A**) THP-1 cells were incubated with IL-1β for 24 h, and miR-1914-5p expression was examined by qRT-PCR. *n =* 3 in each group. * *p* < 0.05 compared with the vehicle group. (**B**) THP-1 cells were transfected with miR-1914-5p inhibitor or negative control (NC) inhibitor for 24 h. miR-1914-5p levels were then assessed by qRT-PCR. An inhibitor of miR-1914-5p significantly suppressed miR-1914-5p accumulation in THP-1 cells compared with the NC. *n =* 3 per group. * *p* < 0.05 compared with the NC transfected group. (**C**) Mac-1 expression in THP-1 cells was examined by flow cytometry after transfection with miR-1914-5p inhibitor or NC for 24 h. THP-1 cells expressing Mac-1 were defined as CD11b^+^, CD18^+^ cells. (**D**) Proportion of cells expressing Mac-1 (CD11b^+^, CD18^+^ cells) among live cells. *n =* 3 in each group. * *p* < 0.05 compared with the NC transfected group. (**E**) Adhesion of THP-1 cells to an EA.hy926 cell monolayer. Fluorescent THP-1 cells and EA.hy926 cells were co-cultured for 1 h after transfection with miR-1914-5p inhibitor or NC for 24 h. Scale bar represents 100 μm. (**F**) The number of fluorescent THP-1 monocytes on the EA.hy926 cell monolayer was determined. *n =* 7 fields in each group. * *p* < 0.05 compared with the NC transfected group.

**Figure 3 ijms-24-02829-f003:**
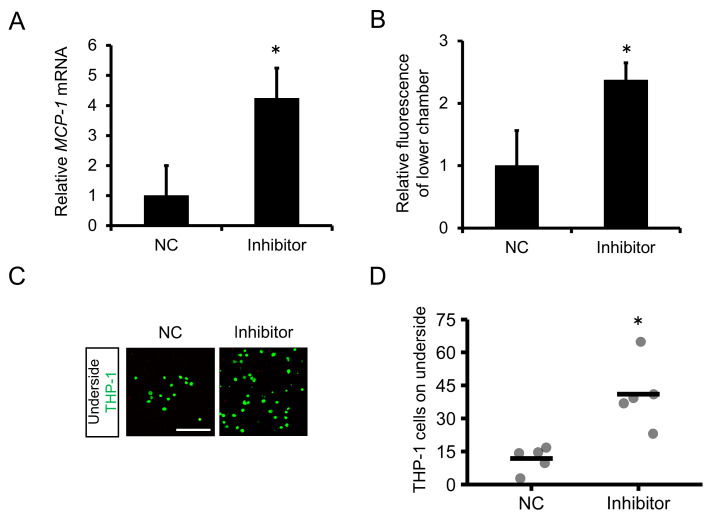
Effect of miR-1914-5p inhibitor on transmigration of monocytes through an endothelial cell layer. (**A**) THP-1 cells were transfected with miR-1914-5p inhibitor or NC for 24 h, and MCP-1 gene expression was examined by qRT-PCR. *n =* 3 in each group. * *p* < 0.05 compared with the NC transfected group. (**B**) Monocyte transmigration through the endothelial cell layer was examined using a transmigration assay after transfection with miR-1914-5p inhibitor or NC for 24 h. Transmigration of THP-1 cells through the endothelial cell layer into the lower chamber was assessed by comparing the relative fluorescence intensity of the upper and lower chambers using a microplate reader. *n =* 3 in each group. * *p* < 0.05 compared with the NC transfected group. (**C**) Fluorescent images of the top and underside of the membrane. THP-1 cells were labeled with the fluorescent dye BCECF-AM (green). Scale bar represents 100 μm. (**D**) The number of fluorescent THP-1 monocytes on the underside of the membrane was determined. *n =* 5 fields in each group. * *p* < 0.05 compared with the NC transfected group.

**Figure 4 ijms-24-02829-f004:**
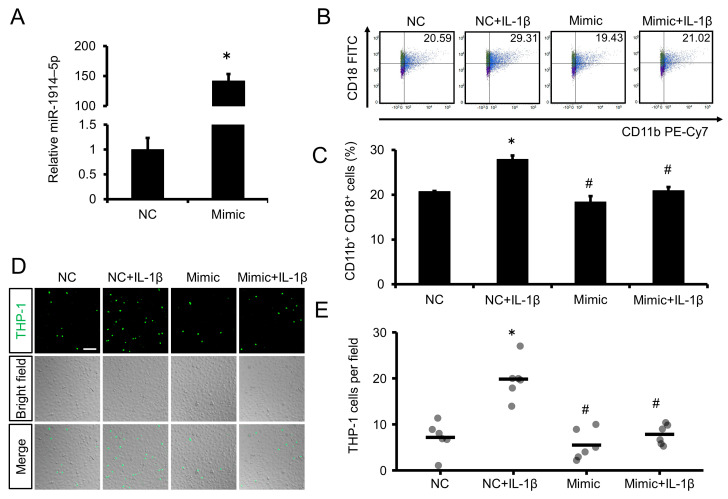
Effect of miR-1914-5p mimic on IL-1β–induced adhesion of monocytes to endothelial cells. (**A**) THP-1 cells were transfected with miR-1914-5p mimic or NC mimic for 24 h. miR-1914-5p levels were then assessed by qRT-PCR. Compared with the NC, the mimic of miR-1914-5p significantly increased miR-1914-5p expression in THP-1 cells. *n =* 3 per group. * *p* < 0.05 compared with the NC transfected group. (**B**) THP-1 cells were transfected with miR-1914-5p mimic or NC for 24 h and then incubated with IL-1β for 24 h. After 24 h of IL-1β treatment, Mac-1 expression in THP-1 cells was examined by flow cytometry. THP-1 cells expressing Mac-1 were defined as CD11b^+^, CD18^+^ cells. (**C**) Proportion of cells expressing Mac-1 (CD11b^+^, CD18^+^ cells) among live cells. *n =* 3 in each group. * *p* < 0.05 compared with the NC group. # *p* < 0.05 compared with the NC+IL-1β group. (**D**) Adhesion of THP-1 cells to an EA.hy926 cell monolayer. THP-1 cells and EA.hy926 cells were co-cultured for 1 h after transfection and IL-1β exposure. Scale bar represents 100 μm. (**E**) The number of fluorescent THP-1 monocytes on the EA.hy926 cell monolayer was determined. *n =* 6 fields in each group. * *p* < 0.05 compared with the NC group. # *p* < 0.05 compared with the NC+IL-1β group.

**Figure 5 ijms-24-02829-f005:**
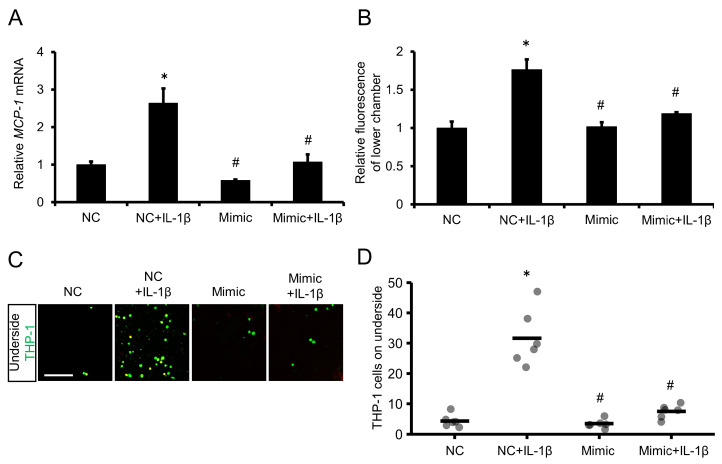
Transfection with miR-1914-5p mimic suppressed IL-1β–induced transmigration of monocytes through an endothelial cell layer. (**A**) THP-1 cells were transfected with miR-1914-5p mimic or NC for 24 h and then incubated with IL-1β for 24 h. MCP-1 gene expression was examined by qRT-PCR. *n =* 3 in each group. * *p* < 0.05 compared with the NC group. # *p* < 0.05 compared with the NC+IL-1β group. (**B**) Monocyte transmigration through an endothelial cell layer was examined using a transmigration assay after transfection and IL-1β exposure. Transmigration of THP-1 cells through the endothelial cell layer into the lower chamber was assessed by comparing the relative fluorescence intensity of the upper and lower chambers using a microplate reader. *n =* 3 in each group. * *p* < 0.05 compared with the NC group. # *p* < 0.05 compared with the NC+IL-1β group. (**C**) Fluorescent images of the top and underside of the membrane. THP-1 cells were labeled with the fluorescent dye BCECF-AM (green). Scale bar represents 100 μm. (**D**) The number of fluorescent THP-1 monocytes on the underside of the membrane was determined. *n =* 6 fields in each group. * *p* < 0.05 compared with the NC group. # *p* < 0.05 compared with the NC+IL-1β group.

## Data Availability

The data presented in this study are available in this article.

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
