# Peer review of "Monocyte-Derived miRNA-1914-5p Attenuates IL-1β–Induced Monocyte Adhesion and Transmigration"

_ijms, 2023, doi:10.3390/ijms24032829_

Round 1

Reviewer 1 Report

The authors conducted a series of experiments to examine whether monocyte miR-1914-5p affects monocyte recruitment in response to IL-1beta. The experiments were well designed and look solid to understand the miR-1914-5p’s role in monocyte for adhesion and transmigration. In particular, the authors find that miR-1914-5p’s role in the upregulation of Mac-1 and MCP-1. On the other hand, the evidence heavily relies on in vitro experiments, and it is not clear whether these results are valid in atherosclerosis in vivo. Additional in vivo experiments to prove the theory are required.

Regarding Figure 1, the data shown were already known. 

Author Response

July 19, 2022

Dr. Kobe Guo, Special Issue Editor

International Journal of Molecular Sciences 

Dear Dr. Guo:

We greatly appreciate your assistance and the instructive review given to our manuscript, entitled “Monocyte-derived miRNA-1914-5p attenuates IL-1β–induced monocyte adhesion and transmigration”.

We have carefully taken the reviewer’s comments into consideration and thoroughly revised the manuscript to address the issues raised by all three reviewers. Revisions in the text are shown in red highlight for additions and red strike-through for deletions. Our detailed, point-by-point responses to the reviewers’ comments are provided in the attached document.

We wish to thank the reviewers for their comments and suggestions for improving the manuscript, as these have resulted in a better and more thorough account of our study. The authors declare that there are no competing interests. All authors have approved the final article. This manuscript has not been submitted previously and is not under consideration for publication by any other journal. We hope that the revisions in the manuscript and our accompanying responses will address all of the issues raised to your full satisfaction and that our manuscript is now suitable for publication in the International Journal of Molecular Sciences.

We look forward to your response.

Yours sincerely,

Mineyoshi Aoyama, Ph.D., M.D.

Department of Pathobiology

Nagoya City University Graduate School of Pharmaceutical Sciences

3-1 Tanabe-dori, Mizuho-ku, Nagoya, Aichi 467-8603, Japan

Phone: +81-52-836-3451; Fax: +81-52-836-3454

E-mail: aomine@phar.nagoya-cu.ac.jp

Manuscript ID: ijms-1767312

Toriuchi et al. “Monocyte-derived miRNA-1914-5p attenuates IL-1β–induced monocyte adhesion and transmigration”

REVIEWER 1

We wish to express our sincere gratitude to the reviewer for these insightful comments, which helped us significantly improve the manuscript.

  1. The experiments were well designed and look solid to understand the miR-1914-5p’s role in monocyte for adhesion and transmigration. In particular, the authors find that miR-1914-5p’s role in the upregulation of Mac-1 and MCP-1. On the other hand, the evidence heavily relies on in vitro experiments, and it is not clear whether these results are valid in atherosclerosis in vivo. Additional in vivo experiments to prove the theory are required.

We thank the reviewer for these comments. We agree with your suggestion, and therefore, we included the following text as a limitation in the Discussion section (p. 9): “Several limitations of our study must be taken into consideration. First, we used an in vitro experimental model to confirm the direct relationship between miR-1914–5p and monocyte adhesion induced by IL-1β in a human monocyte cell line. However, it is not clear whether these results are valid in atherosclerosis in vivo. Further in vivo experiments monitoring the early stages of atherogenesis are needed to confirm these results. In our next study, we plan to examine optimal in vivo experiment models to monitor the early stages of atherogenesis, including apolipoprotein E–deficient mice, low-density lipoprotein receptor–deficient mice, and mice injected with the dominant mutant of proprotein convertase subtilisin/kexin type 9 [31]”

  1. Regarding Figure 1, the data shown were already known.

We thank the reviewer for this comment. We previously reported that miR-1914-5p in endothelial cells suppresses IL-1β–induced ICAM-1 expression and monocyte adhesion to endothelial cells. Many previous studies about adhesion have focused on endothelial cells, not monocytes. Whether IL-1β stimulation could lead to adhesion and transmigration via changes in MAC-1 expression in monocytes remains unclear. Because this issue could be confusing, we included the following text in the Results section (p. 2): “We previously reported that miR-1914-5p in endothelial cells suppresses IL-1β–induced ICAM-1 expression and monocyte adhesion to endothelial cells. However, whether IL-1β stimulation affects monocyte recruitment via changes in adhesion molecule expression in monocytes remains unclear.”

Thank you again for your valuable comments regarding our manuscript. We hope that the revised manuscript is now suitable for publication.

Reviewer 2 Report

The manuscript entitled “Monocyte-derived miRNA-1914-5p attenuates IL-1β-induced monocyte adhesion and transmigration” by Kohki Toriuchi et al, investigates whether monocyte miR-1914-5p affects monocyte recruitment is unclear. In this study, the authors show that IL-1β de-creased miR-1914-5p expression in a human monocyte cell line. Moreover, miR-1914-5p inhibition enhanced adhesion to endothelial cells with the upregulation of macrophage-1 antigen (Mac-1), a counter-ligand to ICAM-1. They also show that the Transmigration through the endothelial layer was also promoted with the upregulation of monocyte chemotactic protein-1 (MCP-1). Furthermore, a miR-1914-5p mimic suppressed IL-1β–induced monocyte adhesion and transmigration in monocytes with Mac-1 and MCP-1 downregulation.

Authors have earlier reported that expression of the miRNA miR-1914-5p is suppressed by IL-1β, an important regulator mediating the expression of ICAM-1 in endothelial cells. However, whether IL-1β affects the expression of miR-1914-5p in monocytes or the regulation of monocyte recruitment via adhesion molecules or chemokine expression remains unclear and the current focus of the study is whether IL-1β suppresses miR-1914-5p expression in monocytes. In addition, we examined the effect of miR-1914-5p on monocyte adhesion and transmigration

The authors have concluded that the miR-1914-5p was identified as an IL-1β–responsive miRNA that suppresses Mac-1 and MCP-1 expression in THP-1 human monocytes. miR-1914-5p inhibitor promoted THP-1 monocyte adhesion and transmigration through the endothelium, but there are queries that need to be addressed.

General comments:

Ø  In fig 1, authors have used various approaches to study effect of IL1b on monocyte adhesion to endothelial transmigration. Bit I wonder there is discrepancy in the study these parameters from within figures. IL1b treatment shows increase in CD11b+ cells. But I wonder how rule out the possibility of neutrophils. Again, the authors used THP1 cells to show the effect of IL1b. Followed by this, the authors again looked for MCP1 expression. How authors made connection here, please explain?

Ø  In Fig 1H, authors performed VE-CAD staining, which is the maker of junctions in endothelial cells, but I wonder how VE-cad expression was observed in THP1 cells. If THP1 cells VE-cadherin, then authors should show VE-cadherin stain also in endothelial cells. As such it doesn’t look like VE-cadherin stain.

Ø  In Fig. 3 authors studied the Effect of miR-1914-5p inhibitor on transmigration of monocytes through an endothelial cell layer, again in fig 3c and 3d, authors VE-cadherin stain in THP1 cells without and with inhibitor, as results indicate that inhibitor treatment increased VE-cadherin expression but here again I would like to ask if authors did see VE-cadherin expression in endothelial cells

Ø  Authors in Fig 4 have used mimic, what is this mimic for? Also, mimic doesn’t have any huge effect in case of fig 4B and C. But in case of fig 4D and E, the mimic shows good effect as compared to NC+IL1b group. Please explain?

Ø  In case of Fig 5C & D, VE-cadherin stain presented in THP1 cells, I would like to see the VE-cadherin stain in endothelial cells as well.?

Author Response

July 19, 2022

Dr. Kobe Guo, Special Issue Editor

International Journal of Molecular Sciences 

Dear Dr. Guo:

We greatly appreciate your assistance and the instructive review given to our manuscript, entitled “Monocyte-derived miRNA-1914-5p attenuates IL-1β–induced monocyte adhesion and transmigration”.

We have carefully taken the reviewer’s comments into consideration and thoroughly revised the manuscript to address the issues raised by all three reviewers. Revisions in the text are shown in red highlight for additions and red strike-through for deletions. Our detailed, point-by-point responses to the reviewers’ comments are provided in the attached document.

We wish to thank the reviewers for their comments and suggestions for improving the manuscript, as these have resulted in a better and more thorough account of our study. The authors declare that there are no competing interests. All authors have approved the final article. This manuscript has not been submitted previously and is not under consideration for publication by any other journal. We hope that the revisions in the manuscript and our accompanying responses will address all of the issues raised to your full satisfaction and that our manuscript is now suitable for publication in the International Journal of Molecular Sciences.

We look forward to your response.

Yours sincerely,

Mineyoshi Aoyama, Ph.D., M.D.

Department of Pathobiology

Nagoya City University Graduate School of Pharmaceutical Sciences

3-1 Tanabe-dori, Mizuho-ku, Nagoya, Aichi 467-8603, Japan

Phone: +81-52-836-3451; Fax: +81-52-836-3454

E-mail: aomine@phar.nagoya-cu.ac.jp

Manuscript ID: ijms-1767312

Toriuchi et al. “Monocyte-derived miRNA-1914-5p attenuates IL-1β–induced monocyte adhesion and transmigration”

REVIEWER 2

We wish to express our sincere gratitude to the reviewer for these insightful comments, which helped us significantly improve the manuscript.

  1. In fig 1, authors have used various approaches to study effect of IL1b on monocyte adhesion to endothelial transmigration. Bit I wonder there is discrepancy in the study these parameters from within figures. IL1b treatment shows increase in CD11b+ cells. But I wonder how rule out the possibility of neutrophils. Again, the authors used THP1 cells to show the effect of IL1b. Followed by this, the authors again looked for MCP1 expression. How authors made connection here, please explain?.

We thank the reviewer for this comment. We included the following text as a limitation in the Discussion section to address this issue (p. 9): “Second, although neutrophils also contribute to plaque formation, we focused on monocytes in our present study [32-34]. miR-1914-5p deficiency–induced interactions between MAC-1 and ICAM-1 could play an important role in neutrophil adhesion to the endothelium. Monocytic MCP-1 could recruit neutrophils to lesions.”

  1. In Fig 1H, authors performed VE-CAD staining, which is the maker of junctions in endothelial cells, but I wonder how VE-cad expression was observed in THP1 cells. If THP1 cells VE-cadherin, then authors should show VE-cadherin stain also in endothelial cells. As such it doesn’t look like VE-cadherin stain.

We thank the reviewer for this comment. VE-CAD staining was performed in endothelial layers to confirm the presence of EA.hy926 endothelial cells. VE-CAD–positive endothelial cells were observed, but the stained image was not clear on the culture membrane shown in Fig. 1H. We were concerned over VE-CAD expression in THP-1 cells by artificial overlapping. To avoid misinterpretation, the image of the top of the membrane was deleted. We also deleted the following text in the Figure legend (p. 3): EA.hy926 cells were stained with anti–VE-cadherin antibody (red).”

  1. In Fig. 3 authors studied the Effect of miR-1914-5p inhibitor on transmigration of monocytes through an endothelial cell layer, again in fig 3c and 3d, authors VE-cadherin stain in THP1 cells without and with inhibitor, as results indicate that inhibitor treatment increased VE-cadherin expression but here again I would like to ask if authors did see VE-cadherin expression in endothelial cells

We thank the reviewer for this comment. VE-CAD staining was performed in endothelial layers to confirm the presence of EA.hy926 endothelial cells. VE-CAD–positive endothelial cells were observed, but the stained image was not clear on the culture membrane shown in Fig. 3C. To avoid misinterpretation, the image of the top of the membrane was deleted. We also deleted the following text in the Figure legend (p. 5): EA.hy926 cells were stained with anti–VE-cadherin antibody (red).”

  1. Authors in Fig 4 have used mimic, what is this mimic for? Also, mimic doesn’t have any huge effect in case of fig 4B and C. But in case of fig 4D and E, the mimic shows good effect as compared to NC+IL1b group. Please explain?

We thank the reviewer for this comment. We overexpressed exogenous miR-1914-5p to confirm its direct effect on monocyte migration. We agree that the change in the percentage of adherent cells was larger than the degree of change in the percentage of monocytes expressing MAC-1. Similar results were observed in the experiment using the miR-1914-5p inhibitor. These results suggest that one or more other adhesion molecules contribute to the effect of miR-1914-5p on monocyte recruitment. We included the following text in the Discussion section to address this issue (p. 9): “Third, we did not consider adhesion factors other than MAC-1. The change in the percentage of adherent cells was larger relative to the amount of change in the percentage of monocytes expressing MAC-1. These results suggest that other adhesion molecules contribute to the effect of miR-1914-5p on monocyte recruitment. Further investigations focusing on lymphocyte function-associated antigen-1, very late antigen-4, and P-selectin glycoprotein ligand-1 on the surface of monocytes in the inflammatory state are needed [35-38].

  1. In case of Fig 5C & D, VE-cadherin stain presented in THP1 cells, I would like to see the VE-cadherin stain in endothelial cells as well.?

We thank the reviewer for this comment. VE-CAD staining was performed in endothelial layers to confirm the presence of EA.hy926 endothelial cells. VE-CAD–positive endothelial cells were observed, but the stained image was not clear on the culture membrane shown in Fig. 5C. To avoid misinterpretation, the image of the top of the membrane was deleted. We also deleted the following text in the Figure legend (p. 7): EA.hy926 cells were stained with anti–VE-cadherin antibody (red).”

Thank you again for your valuable comments regarding our manuscript. We hope that the revised manuscript is now suitable for publication.

Reviewer 3 Report

The manuscript is well written and presented. Data are well interrogated and presented. The hypothesis is well supported by relevant referencing.

One point to note is that miRNA function by targeting signalling/regulatory  hubs. While the reductionist approach is understandable and necessary, comment and discussion should mention other possible putative effects of miRNA-1914-5p, as this miRNA will have a number of targets to monocyte and EC biology and functionality, and possible inflammatory responses/pathways. I would like to see this extrapolated and discussed, backed by references and/or tables.

Author Response

July 19, 2022

Dr. Kobe Guo, Special Issue Editor

International Journal of Molecular Sciences 

Dear Dr. Guo:

We greatly appreciate your assistance and the instructive review given to our manuscript, entitled “Monocyte-derived miRNA-1914-5p attenuates IL-1β–induced monocyte adhesion and transmigration”.

We have carefully taken the reviewer’s comments into consideration and thoroughly revised the manuscript to address the issues raised by all three reviewers. Revisions in the text are shown in red highlight for additions and red strike-through for deletions. Our detailed, point-by-point responses to the reviewers’ comments are provided in the attached document.

We wish to thank the reviewers for their comments and suggestions for improving the manuscript, as these have resulted in a better and more thorough account of our study. The authors declare that there are no competing interests. All authors have approved the final article. This manuscript has not been submitted previously and is not under consideration for publication by any other journal. We hope that the revisions in the manuscript and our accompanying responses will address all of the issues raised to your full satisfaction and that our manuscript is now suitable for publication in the International Journal of Molecular Sciences.

We look forward to your response.

Yours sincerely,

Mineyoshi Aoyama, Ph.D., M.D.

Department of Pathobiology

Nagoya City University Graduate School of Pharmaceutical Sciences

3-1 Tanabe-dori, Mizuho-ku, Nagoya, Aichi 467-8603, Japan

Phone: +81-52-836-3451; Fax: +81-52-836-3454

E-mail: aomine@phar.nagoya-cu.ac.jp

Manuscript ID: ijms-1767312

Toriuchi et al. “Monocyte-derived miRNA-1914-5p attenuates IL-1β–induced monocyte adhesion and transmigration”

REVIEWER 3

We wish to express our sincere gratitude to the reviewer for these insightful comments, which helped us significantly improve the manuscript.

  1. One point to note is that miRNA function by targeting signalling/regulatory hubs. While the reductionist approach is understandable and necessary, comment and discussion should mention other possible putative effects of miRNA-1914-5p, as this miRNA will have a number of targets to monocyte and EC biology and functionality, and possible inflammatory responses/pathways. I would like to see this extrapolated and discussed, backed by references and/or tables.

We thank the reviewer for this comment. We included the following text in the Discussion section to address this issue (p. 8): “miR-1914-5p targets a number of genes in monocytes and affects the biology and functionality of endothelial cells. Using a public database (http://mirdb.org), we previously identified target mRNAs with potential binding sites for miR-1914-5p [17]. Tripartite motif containing 14 (TRIM14), a target gene of miR-1914-5p, was reported to be involved in the expression of endothelial ICAM-1 via activation of nuclear factor–κB (NF-κB) [17,24]. Therefore, miR-1914-5p could be associated with the modulation of NF-κB–mediated inflammatory cytokine production, cell death, and foam cell formation [25]. In addition, zinc finger and BTB domain containing 10 (ZBTB10), another target gene of miR-1914-5p, reportedly downregulates vascular endothelial growth factor (VEGF) [26]. Whether VEGF plays a harmful or beneficial role in atherosclerosis remains controversial [27]. These possibilities must therefore be examined when evaluating therapies targeted to miR-1914-5p.”

Thank you again for your valuable comments regarding our manuscript. We hope that the revised manuscript is now suitable for publication.

Round 2

Reviewer 1 Report

The authors agree with the requirement of in vivo experiments to support their hypothesis but do not conduct the experiments.

Author Response

August 16, 2022

Dr. Kobe Guo, Special Issue Editor

International Journal of Molecular Sciences 

Dear Dr. Guo:

We greatly appreciate your assistance and the instructive review of our manuscript, entitled “Monocyte-derived miRNA-1914-5p attenuates IL-1β–induced monocyte adhesion and transmigration”.

We have carefully taken the reviewer’s comments into consideration and thoroughly revised the manuscript to address the issues raised by all of the reviewers. Revisions in the text are shown in red highlight for additions and red strike-through for deletions. Our detailed, point-by-point responses to the reviewers’ comments are provided in the attached document.

We wish to thank the reviewers for their comments and suggestions for improving the manuscript, as these have resulted in a better and more thorough account of our study. The authors declare that there are no competing interests. All authors have approved the final article. This manuscript has not been submitted previously and is not under consideration for publication by any other journal. We hope that the revisions in the manuscript and our accompanying responses address all of the issues raised to your full satisfaction and that our manuscript is now suitable for publication in the International Journal of Molecular Sciences.

We look forward to your response.

Sincerely,

Mineyoshi Aoyama, Ph.D., M.D.

Department of Pathobiology

Nagoya City University Graduate School of Pharmaceutical Sciences

3-1 Tanabe-dori, Mizuho-ku, Nagoya, Aichi 467-8603, Japan

Phone: +81-52-836-3451; Fax: +81-52-836-3454

E-mail: aomine@phar.nagoya-cu.ac.jp

Manuscript ID: ijms-1767312

Toriuchi et al. “Monocyte-derived miRNA-1914-5p attenuates IL-1β–induced monocyte adhesion and transmigration”

REVIEWER 1

We wish to express our sincere gratitude to the reviewer for these insightful comments, which helped us significantly improve the manuscript.

  1. The authors agree with the requirement of in vivo experiments to support their hypothesis but do not conduct the experiments.

We thank the reviewer for this comment. We attempted to create an in vivo model, but unfortunately, we need additional time to establish appropriately stable experiments and perform reliable analyses. For now, we can only report in vitro cellular analyses. Due to some overstatements regarding the in vitro experiments only, we deleted the following text in the Discussion section (p. 9): “These findings suggest that promoting miR-1914-5p expression could be an effective new therapeutic approach for treating atherosclerosis. Furthermore, our results suggest that monitoring miR-1914-5p expression in monocytes could be a useful novel diagnostic marker for atherosclerosis.” We replaced the deleted text with the following (p. 9): “These findings suggest that miR-1914-5p in monocytes regulates monocyte recruitment under inflammatory conditions through vascular endothelial cells. Further investigation using an in vivo pathological model could lead to an effective new therapeutic approach that promotes miR-1914-5p expression to treat atherosclerosis.“

We also added the following text regarding an additional experimental plan (p. 9): “To determine the contribution of monocytic miR-1914-5p to atherogenesis, we would isolate monocytes from wild-type mice and generate monocytes exhibiting lower miR-1914-5p expression using an inhibitor of transfection and generate monocytes exhibiting high miR-1914-5p expression via mimic transfection. In addition, we would intravenously inject each transfectant into experimental model mice.“

Thank you again for your valuable comments regarding our manuscript. We hope that the revised manuscript is now suitable for publication.

Reviewer 2 Report

The Revised manuscript entitled “Monocyte-derived miRNA-1914-5p attenuates IL-1β-induced monocyte adhesion and transmigration” by Kohki Toriuchi et al, investigates whether monocyte miR-1914-5p affects monocyte recruitment is unclear. In this study, the authors show that IL-1β de-creased miR-1914-5p expression in a human monocyte cell line. Moreover, miR-1914-5p inhibition enhanced adhesion to endothelial cells with the upregulation of macrophage-1 antigen (Mac-1), a counter-ligand to ICAM-1. They also show that the Transmigration through the endothelial layer was also promoted with the upregulation of monocyte chemotactic protein-1 (MCP-1).

The authors have concluded that the miR-1914-5p was identified as an IL-1β–responsive miRNA that suppresses Mac-1 and MCP-1 expression in THP-1 human monocytes. miR-1914-5p inhibitor promoted THP-1 monocyte adhesion and transmigration through the endothelium, but there are queries that need to be addressed.

General comments:

Ø  Authors have addressed the points and concerns raised to the large extent.

Ø  It would be better if authors would make scatter plots rather than bar graphs where ever required

Ø  I would suggest again read the manuscript properly and look for any language errors and plagiarism.

Author Response

August 16, 2022

Dr. Kobe Guo, Special Issue Editor

International Journal of Molecular Sciences 

Dear Dr. Guo:

We greatly appreciate your assistance and the instructive review of our manuscript, entitled “Monocyte-derived miRNA-1914-5p attenuates IL-1β–induced monocyte adhesion and transmigration”.

We have carefully taken the reviewer’s comments into consideration and thoroughly revised the manuscript to address the issues raised by all of the reviewers. Revisions in the text are shown in red highlight for additions and red strike-through for deletions. Our detailed, point-by-point responses to the reviewers’ comments are provided in the attached document.

We wish to thank the reviewers for their comments and suggestions for improving the manuscript, as these have resulted in a better and more thorough account of our study. The authors declare that there are no competing interests. All authors have approved the final article. This manuscript has not been submitted previously and is not under consideration for publication by any other journal. We hope that the revisions in the manuscript and our accompanying responses address all of the issues raised to your full satisfaction and that our manuscript is now suitable for publication in the International Journal of Molecular Sciences.

We look forward to your response.

Sincerely,

Mineyoshi Aoyama, Ph.D., M.D.

Department of Pathobiology

Nagoya City University Graduate School of Pharmaceutical Sciences

3-1 Tanabe-dori, Mizuho-ku, Nagoya, Aichi 467-8603, Japan

Phone: +81-52-836-3451; Fax: +81-52-836-3454

E-mail: aomine@phar.nagoya-cu.ac.jp

Manuscript ID: ijms-1767312

Toriuchi et al. “Monocyte-derived miRNA-1914-5p attenuates IL-1β–induced monocyte adhesion and transmigration”

REVIEWER 2

We wish to express our sincere gratitude to the reviewer for these insightful comments, which helped us significantly improve the manuscript.

  1. It would be better if authors would make scatter plots rather than bar graphs where ever required

We thank the reviewer for this comment. We converted Figs. 1D, 1I, 2F, 3D, 4E, and 5D from bar graphs to dot plots to enhance clarity and understanding. We also checked and corrected the n numbers in the figure legends (pp. 3-5).

  1. I would suggest again read the manuscript properly and look for any language errors and plagiarism.

We thank the reviewer for this comment. We again checked for plagiarism. The manuscript was re-checked for plagiarism and proofread by an English-speaking, experienced scientific editor.

Thank you again for your valuable comments regarding our manuscript. We hope that the revised manuscript is now suitable for publication.

Round 3

Reviewer 1 Report

The authors should conduct further research to prove that the results presented here are valid in animal models.

Author Response

September 16, 2022

Dr. Kobe Guo, Special Issue Editor

International Journal of Molecular Sciences 

Dear Dr. Guo:

We greatly appreciate your assistance and the instructive review given to our manuscript, entitled “Monocyte-derived miRNA-1914-5p attenuates IL-1β–induced monocyte adhesion and transmigration”.

We have carefully taken the editor’s and reviewer’s comments into consideration and thoroughly revised the manuscript to address the issues raised by all three reviewers. Revisions in the text are shown in red highlight for additions and red strike-through for deletions. Our detailed, point-by-point responses to the reviewers’ comments are provided in the attached document.

We wish to thank the reviewers for their comments and suggestions for improving the manuscript, as these have resulted in a better and more thorough account of our study. The authors declare that there are no competing interests. All authors have approved the final article. This manuscript has not been submitted previously and is not under consideration for publication by any other journal. We hope that the revisions in the manuscript and our accompanying responses will address all of the issues raised to your full satisfaction and that our manuscript is now suitable for publication in the International Journal of Molecular Sciences.

We look forward to your response.

Yours sincerely,

Mineyoshi Aoyama, Ph.D., M.D.

Department of Pathobiology

Nagoya City University Graduate School of Pharmaceutical Sciences

3-1 Tanabe-dori, Mizuho-ku, Nagoya, Aichi 467-8603, Japan

Phone: +81-52-836-3451; Fax: +81-52-836-3454

E-mail: aomine@phar.nagoya-cu.ac.jp

Manuscript ID: ijms-1767312

Toriuchi et al. “Monocyte-derived miRNA-1914-5p attenuates IL-1β–induced monocyte adhesion and transmigration”

REVIEWER 1

We wish to express our sincere gratitude to the reviewer for these insightful comments, which helped us significantly improve the manuscript.

  1. The authors should conduct further research to prove that the results presented here are valid in animal models.

We thank the reviewer for this comment. Ideally, in vivo experiments should be conducted. Based on the editor’s comment, we have checked whether miRNA-1914-5p is conserved in mice. Unfortunately, miR-1914-5p expression has been confirmed only in humans and not in rodents. As such, rodent models are not currently available. We deleted the following text in the Discussion section (p. 9): “In our next study, we plan to examine optimal in vivo experiment models need to be used to monitor the early stages of atherogenesis, including apolipoprotein E–deficient mice, low-density lipoprotein receptor–deficient mice, and mice injected with the dominant mutant of proprotein convertase subtilisin/kexin type 9 [31]. To determine the contribution of monocytic miR-1914-5p to atherogenesis, we would isolate monocytes from wild-type mice and generate monocytes exhibiting lower miR-1914-5p expression using an inhibitor transfection and generate monocytes exhibiting high miR-1914-5p expression via mimic transfection. In addition, we would intravenously inject each transfectant into experimental model mice. We replaced the deleted text with the following (p. 9):Further in vivo experiments monitoring the early stages of atherogenesis are needed to confirm these in vitro results, including apolipoprotein E–deficient mice, low-density lipoprotein receptor–deficient mice, and mice injected with the dominant mutant of proprotein convertase subtilisin/kexin type 9 [31]. Unfortunately, these rodent models are not currently available, because miR-1914-5p expression has been confirmed only in human and not in rodents.

Thank you again for your valuable comments regarding our manuscript. We hope that the revised manuscript is now suitable for publication.
